



# Composition and mixing state of Arctic aerosol and cloud residual particles from long-term single-particle observations at Zeppelin Observatory, Svalbard

Kouji Adachi[1], Yutaka Tobo[2,3], Makoto Koike[4], Gabriel Freitas[5], Paul Zieger[5], Radovan Krejci[5]

[1] Department of Atmosphere, Ocean, and Earth System Modeling Research, Meteorological Research Institute, Tsukuba, 3050052, Japan
[2] National Institute of Polar Research, Tachikawa, 1908518, Japan
[3] Department of Polar Science, School of Multidisciplinary Sciences, The Graduate University for Advanced Studies,
SOKENDAI, Tachikawa, 1908518, Japan
[4] Department of Earth and Planetary Science, Graduate School of Science, The University of Tokyo, Tokyo, 1130033, Japan
[5] Department of Environmental Science & Bolin Centre for Climate Research, Stockholm University, Stockholm, 10691, Sweden

*Correspondence to*: Kouji Adachi (adachik@mri-jma.go.jp)

**Abstract.** The Arctic region is sensitive to climate change and is warming faster than the global average. Aerosol particles change cloud properties by acting as cloud condensation nuclei and ice nucleating particles, thus influencing the Arctic climate system. Therefore, understanding the aerosol particle properties in the Arctic is needed to interpret and simulate their influences on climate. In this study, we collected ambient aerosol particles using whole-air and $PM_{10}$ inlets and residual particles of cloud
droplets and ice crystals from Arctic low-level clouds (typically, all-liquid or mixed-phase clouds) using a counterflow virtual impactor inlet at the Zeppelin Observatory near Ny-Ålesund, Svalbard, within a time frame of 4 years. We measured the composition and mixing state of individual fine-mode particles using transmission electron microscopy. On the basis of their composition, the aerosol and cloud residual particles were classified into mineral dust, sea salt, K-bearing, sulfate, and carbonaceous particles. The number fraction of aerosol particles showed seasonal changes, with sulfate dominating in summer
and sea salt increasing in winter. There was no measurable difference in the fractions between ambient aerosol and cloud residual particles collected at ambient temperatures above 0°C. On the other hand, cloud residual samples collected at ambient temperatures below 0°C had several times more sea salt and mineral dust particles and fewer sulfates than ambient aerosol samples, suggesting that sea spray and mineral dust particles may influence the formation of cloud particles in Arctic mixed-phase clouds. We also found that 43% of mineral dust particles from cloud residual samples were mixed with sea salt, whereas
only 18% of mineral dust particles in ambient aerosol samples were mixed with sea salt. This study highlights the variety of aerosol compositions and mixing states that influence or are influenced by aerosol-cloud interactions in Arctic low-level clouds.

## 1 Introduction

Aerosol particles play important and multiple roles in the Arctic climate system (Schmale et al., 2021). They affect solar radiation by scattering and absorbing it and act as nuclei of cloud droplets or ice crystals. Deposition of light-absorbing particles,
such as mineral dust and black carbon, reduces the surface albedo of snow and ice and accelerates their melting (Aoki et al., 2014). Information on aerosol properties such as compositions, size distributions, and atmospheric mass and number concentrations are widely monitored over the Arctic to understand their climatological contributions (e.g., Abbatt et al., 2019; Platt et al., 2022). Information on individual particle properties is also important to understand the fundamental chemical and physical processes of aerosol particles (e.g., Laskin et al., 2016; Buseck and Pósfai, 1999; Pósfai and Buseck, 2010; Adachi et
al., 2010; Krieger et al., 2012). Such individual particle properties include (1) particle shape that affects light scattering and



absorption, (2) mixing state that indicates how different components mix within individual particles, and (3) compositions and structures of particle surfaces.

Microscopic analysis with a spectrometer reveals physical and chemical properties of individual particles, e.g., transmission and scanning electron microscopy coupled with an energy-dispersive X-ray spectrometer (TEM-EDS and SEM-EDS, respectively) and scanning transmission X-ray microscopy with near-edge X-ray absorption fine structure spectroscopy (Laskin et al., 2016; Li et al., 2016). These techniques have been applied to various Arctic aerosol particles, including those around Ny-Ålesund, the current observation site, using samples collected during intensive observation campaigns (Weinbruch et al., 2012; Moroni et al., 2017; Moroni et al., 2020; Behrenfeldt et al., 2008; Gen et al., 2010; Chi et al., 2015; Anderson et al., 1992; Hara et al., 2003).

In the Arctic, there is a distinct seasonality in the concentration, composition, and sources of aerosol particles, primarily due to changes in available sunlight and air mass transport pathways (Tunved et al., 2013). Changes in available sunlight subsequently alter snow and sea ice coverages and affect aerosol emissions such as mineral dust and sea salt aerosol particles (Willis et al., 2018; Schmale et al., 2021; Schmeisser et al., 2018). Seasonal changes in available sunlight also determine the photochemistry and halogen cycles of aerosol particles (Arnold et al., 2016; Shaw, 1995; Willis et al., 2018). The Arctic atmosphere is heavily influenced by anthropogenic emissions from low- and mid-latitudes during winter and early spring. In contrast, the influence of long-range transport is weakened in summer. Thus, observations covering all seasons are needed to determine the influence of Arctic aerosol particles on climate.

Cloud phases (liquid, ice, and mixed phases) and sizes and concentrations of cloud droplets affect precipitation and solar radiation, influencing the air temperature (Cesana and Storelvmo, 2017; Baker, 1997). Cloud properties are, besides the governing meteorology, also influenced by aerosol particles that form cloud droplets (Fan et al., 2016). The hygroscopicity (i.e., particle composition and mixing state) and size of aerosol particles determine whether they can become cloud condensation nuclei (CCN) under certain ambient supersaturated conditions (Andreae and Rosenfeld, 2008). Conversely, the ability to become ice nucleating particles (INPs) is more complex than that of CCNs. In general, only one of the tens to thousands of all particles can act as INP (Kanji et al., 2017). For example, Koike et al. (2019) showed that above 0°C, the cloud droplet number concentration is simply controlled by the number concentration of aerosol particles larger than 70 nm. In contrast, below 0°C, the relationship becomes obscure because of the activation of heterogeneous ice nucleation involving a small fraction of aerosol particles. It is highly uncertain which particles become INPs, but solid particles with characteristic surface structures or compositions (e.g., feldspar and biogenic/organic materials) may be more likely to become INPs at higher temperatures than others (Atkinson et al., 2013; Creamean et al., 2013; Kanji et al., 2017).

Several techniques have been applied to measure the number concentration of INPs, including online continuous flow diffusion chambers and offline filter measurements (DeMott et al., 2017; Tobo et al., 2019). These methods measure the number of INPs by exposing samples to supersaturation or supercooled water droplets below 0°C in a chamber or cell. Meanwhile, measurements using a counterflow virtual impactor (CVI) inlet directly observe ambient clouds and collect cloud residual particles (Noone et al., 1988; Shingler et al., 2012). CVI inlets have been used in mountain sites and aircraft-based observations (Kamphus et al., 2010; Karlsson et al., 2021; Cziczo et al., 2004; Heintzenberg et al., 1996; Cziczo and Froyd, 2014).

The observations for this study were conducted at the Zeppelin Observatory near Ny-Ålesund, Svalbard, Norway. The observation site is designed to monitor the Arctic background air and has been operating since 1989 (Platt et al., 2021; Lee et al., 2020; Tunved et al., 2013; Karl et al., 2019; Dekhtyareva et al., 2018; Karlsson et al., 2021). Additionally, continuous CVI sampling of cloud particles has been conducted since November 2015 (Karlsson et al., 2021).

This study is based on aerosol particles and cloud residual particles collected from the Zeppelin Observatory from 2017 to 2021 using impactor samplers and the CVI inlet. This period overlapped with the campaigns of the Ny-Ålesund AeroSol Cloud ExperimeNT (NASCENT) (Pasquier et al., 2022) and the Multidisciplinary drifting Observatory for the Study of Arctic Climate (MOSAiC) (Shupe et al., 2022). We focused on measuring individual particles using scanning TEM coupled with


EDS (STEM-EDS). Previous studies have measured individual particle compositions at Ny-Ålesund using samples collected
during short-term campaigns. Instead, this study is the first attempt to provide a multiyear picture of aerosol and cloud residual
particle properties and variability. We aim to determine the composition, mixing state, and number fraction of ambient aerosol
and cloud residual particles during different seasons. We also explore the possibility of INP characterization from cloud
residual particles collected below 0℃. Overall, we report observations on individual aerosol and cloud residual particles that
provide essential information for understanding aerosol–cloud interactions and the influence of natural and anthropogenic
sources on the Arctic climate.

## 2 Methods

### 2.1 Sampling

Sampling was conducted at the Zeppelin Observatory (78°54′ N, 11°53′ E) near Ny-Ålesund, Svalbard, Norway (Fig. 1a). The
observatory is located on the ridge of Mt. Zeppelin (Fig. 1b), with an inlet height of 480 m a.s.l. (Platt et al., 2022). The station
is representative of Arctic background conditions. The observatory is often covered by low-level clouds, making it suitable for
studying aerosol-cloud interactions in the Arctic (Dekhtyareva et al., 2018; Platt et al., 2022; Koike et al., 2019; Turnved et
al., 2013, Karlsson et al., 2021).

TEM samples were collected through three inlets (see Section 2.2 for inlet details) at the Zeppelin Observatory (Figs. 1c and
S1) (Koike et al., 2019; Karlsson et al., 2021) using impactor samplers (AS-16W and AS-24W, Arios Inc., Tokyo, Japan). The
samplers have two impactor stages for fine and coarse mode aerosol particles. The fine mode stage has lower and upper 50%
cutoff sizes with aerodynamic diameters of 0.1 and 0.7 µm, respectively, with a flow rate of 1.0 L m$^{-1}$, and the coarse mode
stage collects particles larger than 0.7 µm. This study focused on the fine-mode samples. The AS-16W and AS-24W samplers
have the same sampling conditions, except that the former has 16 TEM samples preset and the latter has 24 TEM samples
preset. The samplers are operated using a built-in timer or a computer that controls the sampling. The samplers have been used
in various environments in our previous measurements (Adachi et al., 2019; 2020; 2021; 2022a). All aerosol and cloud residual
particles were collected on a 200-mesh Cu grid with a formvar carbon substrate (U1007, EM-Japan, Tokyo, Japan). Sampling
times were mostly 30 or 60 min, depending on the concentration of atmospheric particles. The sampled air was not actively
dried, but because of the high-temperatures differences between the ambient air and the laboratory with the samplers, the
sampled air had a low relative humidity (RH) when collected (mean RH value: 13 ± 7%; Karlsson et al., 2021).

### 2.2 Sampling categories based on inlets

TEM samples were classified into two subsets based on sampling type: (1) ambient aerosol samples and (2) cloud residual
samples (Fig. S1). The sampling periods, sample numbers, and temperature range are shown in Table 1.

**Ambient aerosol samples**: A PM$_{10}$ inlet and whole-air inlet were used to collect ambient aerosol samples. Aerosol particles
passing through the PM$_{10}$ inlet were collected intensively in eight campaign periods ranging from 3 days to 3 weeks (Table 1).
Samples were collected at preset intervals (e.g., 1, 3, or 6 h, depending on campaigns) using a built-in timer. A splitter was
used to split and share sampled air with other instruments, such as an aerosol sampler for INP measurement and a filter-based
absorption photometer (Ohata et al., 2019, 2021). Ambient aerosol samples were also collected through a whole-air inlet
(Karlsson et al., 2021). The samplings were performed manually with a remote-control during cloud-free periods by the same
TEM sampler used for cloud residual particle sampling. Sampling was timed to occur before and after cloud periods or during
periods of interest based on online observation data (e.g., meteorology and particle concentrations). Sampling times were
arbitrary; however, the ambient aerosol samples covered all months during the sampling periods except June.

In this study, the samples from the PM$_{10}$ and whole-air inlets are treated as ambient aerosol samples because they are essentially
the same sample set and showed similar results. Although the upper particle sizes of these inlets are different, the difference





in cutoff sizes of these inlets does not affect the results because the samplers have cutoff sizes (0.7 µm) that are much smaller

than the cutoff sizes of these inlets (10 µm).

**Cloud residual samples**: Cloud residual samples were collected through a ground-based CVI inlet (Brechtel Manufacturing Inc., USA, Model 1205) (Karlsson et al., 2021). The CVI was activated during cloud periods of < 1 km visibility as measured using a visibility sensor. Cloud residual particles were remaining from dried cloud droplets or ice crystals after passing the CVI. These cloud residual samples were classified into three categories based on ambient air temperature: above 0°C, 0°C to

−4°C, and below −4°C. All cloud residual particles above 0°C are liquid droplets, whereas those below 0°C are supercooled water droplets or ice crystals. Note that cloud droplets may have undergone colder temperatures than the sampling site, and the measured temperature may not be the same as the activated temperature for forming ice crystals (Carlsen and David, 2022).

### 2.3 CVI inlet

A CVI uses a counter flow toward the sample flow so that only large particles, i.e., cloud droplets and ice crystals, can pass

through the impactor (Noone et al., 1988). Interstitial aerosol particles are not collected because they are transported along the streamline away from the impactor. In this study, the lower 50% cutoff size for cloud droplets and ice crystals is 6 to 7 µm aerodynamic diameter. Those larger than 40 µm are also lost because of the long evaporation time at the inlet (Karlsson et al., 2021). Cloud droplets and ice crystals are dried in dry counterflow air with a dew point of −40°C at the inlet. For further information on the CVI inlet used in this study, see Karlsson et al. (2021). Technical descriptions can be found in Shingler et

al. (2012).

### 2.4 TEM analyses

All TEM samples were analyzed using the TEM and STEM modes of a transmission electron microscope (JEM-1400, JEOL, Japan) with an energy-dispersive X-ray spectrometer (EDS; X-Max 80 mm, Oxford instrument, Japan) (Adachi et al., 2022b). Particle composition and size (area equivalent diameter) were analyzed using STEM-EDS measurement at an acceleration

voltage of 120 kV. TEM images were taken before and after STEM-EDS measurements. Twenty elements (C, N, O, Na, Mg, Al, Si, P, S, Cl, K, Ca, Ti, V, Cr, Mn, Fe, Ni, Zn, and I) were selected and measured in STEM-EDS measurements with an acquisition time of 20 s. The detection limits of measured weight % were defined as the peak count intensity being less than the sigma value of the peak area and were generally <0.1 wt%. For the STEM-EDS measurements, the minimum number of pixels for each particle at a magnification of 6000× in the STEM image was set to 100, resulting in the minimum particle size

of 0.26 µm in area equivalent diameter. All particles larger than the cutoff size and having a stronger contrast than the background substrate in the analyzed area were measured. The area of each particle was determined by making a binary image from the histogram of image contrasts using a threshold value to distinguish between particles and the substrate in the STEM image (Adachi et al., 2019). Note that for particles with a flat shape or low density, the area-equivalent diameter may be larger than the aerodynamic diameter because of differences in the definition of particle size. On average, approximately 170 particles

were analyzed from each TEM sample, yielding 40,790 particles from 239 TEM samples.

### 2.5 Particle classification based on the compositions

The particles measured using STEM-EDS were classified into six categories based on key elements in the following aerosol species: mineral dust particle (Al > 1 wt% and Fe > 1 wt%), sea salt particle (Na > 1 wt%), K-bearing particle (K > 1 wt%), sulfate particle (S > 1 wt%), carbonaceous particle (C + O > 90wt%), and others (Fig. S2). Secondary aerosol species (e.g.,

sulfate and carbonaceous particles) easily condense on preexisting primary particles or aggregate with other particles. Thus, many particles are a mixture of several components (e.g., sea salt + sulfate) (Figs. 2-6). The classification followed by the flow chart (Fig. S2) is sensitive to counting primary particles (mineral dust, sea salt, and K-bearing particles) and likely


underestimates secondary aerosol species (e.g., sulfate and carbonaceous particles) when compared with a measurement based on mass fractions.

## 3 Results and discussions

### 3.1 Particle characterizations measured using TEM

The following sections describe the general characteristics of major aerosol species (mineral dust, sea salt, K-bearing, sulfate, and carbonaceous particles) based on TEM measurements. Additionally, the TEM results of iodine-bearing particles are shown in the Supplementary text (Fig. S3) in consideration of their importance for new particle formation in the Arctic (Baccarini et al., 2020; Allan et al., 2015; Sipilä et al., 2016).

### 3.1.1 Mineral dust particles

Mineral dust particles originate from local sources in the Arctic (e.g., glacial outwash plains) and distant sources in low- and mid-latitudes (Tobo et al., 2019; Song et al., 2021; Moroni et al., 2016; Shi et al., 2022). Mineral dust particles detected in this study are primarily aluminosilicates (mainly Si and Al with Na, K, Ca, and others) aggregated with other mineral compounds containing Si, Fe, and Ca (Figs. 2-4a). They also have a relatively high amount of Na, Mg, S, and Cl, originating from sea salt and sulfate attached to mineral dust particles (Fig. 2). As a result, they primarily consist of Al, Si, Fe, and O with minor amounts of other elements such as Na, Mg, S, Cl, K, and Ca (Table S1). They present crystal structures and irregular shapes and are relatively large (with an average area equivalent diameter of 0.56 µm) compared with other species (Fig. 4a).

### 3.1.2 Sea salt particles

Sea salt particles originate from sea spray from the open sea and local leads, frost flowers, and blowing snow in the Arctic (Domine et al., 2004; Hara et al., 2017). Sea salt particles are composed mainly of NaCl in their fresh state, with others from seawater components such as Mg- and Ca-bearing salts (Fig. 4b). Sea salt particles with relatively small sizes (typically < 1 µm) commonly occurred on the substrate in a spherical form. The spherical shape indicates that they were hydrated at the time of collection (Fig. 3). Sea salt particles readily react with acidic substances (e.g., sulfate, nitrate, and organics) (Chi et al., 2015; Geng et al., 2010; Hara et al., 2002) to form sodium sulfate or nitrate, with Cl being released as HCl. In this study, reactions between sea salt and sulfate are more active from spring to fall, i.e., individual sea salt particles are high in S and low in Cl (Fig. S4), possibly due to the abundance of sulfuric acid in the atmosphere (Beck et al., 2021). In contrast, many sea salt particles retain Cl in winter. Large sea salt particles (> 1 µm) sometimes have Ca and Mg as sulfate or chloride forms around them (Fig. 4b). Since both $CaCl_2$ and $MgCl_2$ have low deliquescence RH values (<35% at 25℃) (Guo et al., 2019), they could have deliquesced prior to NaCl and showed phase separations. Salter et al., (2016) showed that Ca enriches in sea spray particles with < 0.1µm in aerodynamic diameter, which is a lower limit of our samples, and thus we did not observe the Ca enrichment. Nitrate coatings were not observed in this study, although reactions between sea salt particles and nitrate have been reported at the Ny-Ålesund ground site in previous studies (Chi et al., 2015; Geng et al., 2010). Possible reasons are given in the Supplementary text.

### 3.1.3 K-bearing particles

K-bearing particles contain S, indicating that they are mostly potassium sulfate (Fig. 5). Biomass and biofuel burnings are known to be their major sources (Li et al., 2003; Andreae, 2019). Soot particles emitted from biomass burning are found in K-bearing particles (red arrows in Fig. 5b). When K is emitted from biomass burning, it takes sulfate or chloride forms. The latter reacts with sulfate in the smoke plume, forming potassium sulfates (Li et al., 2003). During TEM measurements, they were





easily damaged because of exposure to electron beams (Fig. 5), similar to ammonium sulfate or ammonium bisulfate (Adachi et al., 2014).

### 3.1.4 Sulfate particles

In the Arctic, sulfates mainly originate from anthropogenic and natural (marine) sources (Udisti et al., 2016). The contributions of anthropogenic sources peak in spring, whereas those of marine sources increase in summer (Platt et al., 2022; Lee et al.,

2020; Udisti et al., 2016). Sulfate particles from spring and summer present spherical or cocoon-like shapes (Figs. 6a and 6b), similar to those from other areas (Ueda et al., 2021). Some sulfate particles also have satellite structures (rings of small spots around the sulfate particles; Fig. 6b), indicating that they are less viscose at the time of collection (Kojima et al., 2004).

### 3.1.5 Carbonaceous particles

Carbonaceous particles include secondary and primary organic aerosols and soot particles. Of these, secondary organic aerosol

particles dominate this category and have a weaker contrast in bright-field TEM images (gray particles or coatings) than sulfates (Fig. 6c). Particles with an amorphous structure (e.g., secondary organic aerosol particles) scatter electron beam less than those with a crystal structure (e.g., sulfate, mineral dust, and sea salt), resulting in a weaker contrast of organics in the TEM image. These secondary organic materials often have sulfate inclusions (sulfate cores in Fig. 6c). Primary organic particles, such as tarballs from biomass burning, have been observed in the Arctic atmosphere (Adachi et al., 2021; Moroni et

al., 2017, 2020) but were not found in this study. Most of the soot particles were coated with sulfates (e.g., ammonium sulfate and potassium sulfate), and the number of bare (externally mixed) soot particles was small (<0.1% of this category). Note that the volatile or semivolatile organic fractions are lost during sampling, storage, and TEM measurements.

Previous studies suggest that biological materials attached to or emitted with sea spray and mineral dust particles may contribute to INP activity in high latitudes (Tobo et al., 2019; Hartmann et al., 2020; Rinaldi et al., 2021; Creamean et al.,

2022; Xi et al., 2022; Yun et al., 2022). Although TEM can detect such primary biological particles if they are present (Adachi et al., 2020), no such biological materials were found in the current fine mode particles. Further studies on particles larger than ~1 µm in aerodynamic diameter are required, as biological particles are generally abundant in coarse mode particles.

### 3.1.6 Others

This category includes particles enriched in Fe, Si, and Ca. Additionally, artifact particles, such as stainless particles detached

from the CVI surface due to collision with ice crystals, have been reported in previous CVI studies (Cziczo et al., 2004) and can be included in this category. However, the cloud residual samples contained only a few stainless particles characterized by Fe and Cr, and their influence was negligible in our samples. The overall number fraction of this category is small (<1%; Figs. 7 and S5) and will not be discussed further in this study.

### 3.2 Number fractions of ambient aerosol and cloud residual particles and their seasonality

The number fractions and size distributions for each particle category are shown for ambient aerosol and cloud residual samples (Fig. 7). Sulfate and sea salt particles had the first- and second-largest number fractions. Together, they accounted for >80% of all measured particles (Figs. 7 and S5). The fraction of sea salt particles increased with increasing particle size. In contrast, the sulfate particle fraction increased with decreasing particle size for both ambient aerosol and cloud residual samples (Fig. 7). Carbonaceous particle fraction increased with decreasing particle size, whereas no clear size dependence was observed for

K-bearing particle fraction. Mineral dust particle fractions were small on average (0.4% and 1.1% for ambient aerosol and cloud residual samples, respectively) and generally increased in large-size bins.

The abundance ratios of aerosol particles in different size bins were roughly consistent with previous measurements in the Arctic (Kirpes et al., 2018; Abbatt et al., 2019; Adachi et al., 2021). For example, Weinbruch et al. (2012) analyzed aerosol





particles collected at the Zeppelin Observatory between July 2007 and December 2008 using SEM. They found that sea salt
particles were most abundant in particles larger than 0.5 µm and that secondary aerosol particles (mixtures of sulfate, nitrate,
and organic material) were dominant in particles smaller than 0.5 µm.

TEM samples were classified into four seasonal groups based on the month of collection: spring (March to May), summer
(June to August), fall (September to November), and winter (December to February) (Figs. 8 and S5). Summer and winter
mostly cover polar day and night periods, respectively, and spring and fall are the intermediate periods. Arctic haze builds up
in winter and peaks in spring, whereas aerosol concentrations are the lowest in fall (Tunved et al., 2013).

Mineral dust particle fractions were generally small (<1%) for all seasons and all sampling categories, except for winter cloud
residual samples (8%) (Fig. 8h). The sea salt fraction was relatively high in the winter ambient aerosol and winter cloud
residual samples (28% and 64%, respectively) and low in the summer ambient aerosol samples (7%). The K-bearing particle
fraction was relatively high in the spring samples. The sulfate fraction was relatively high in summer for samples from both
sampling categories (66%–81%). The carbonaceous particle fraction showed no clear seasonal trends or differences between
sampling categories.

These seasonal changes in the aerosol number fraction of ambient aerosol samples were generally consistent with other bulk
observations of aerosol mass concentrations in the Arctic areas (e.g., Alert, Canada; Leaitch et al., 2018). For example, the
mass concentrations of relatively large (accumulation mode) sulfate aerosol particles were high in the spring when long-range
transport of anthropogenic pollutants and photochemical processes increased, whereas those smaller than 100 nm increased in
summer when marine biogenic emissions were high (Croft et al., 2016; Willis et al., 2018; Leaitch et al., 2018; Udisti et al.,
2016). Meanwhile, the sea salt fraction was high in winter and spring when strong winds produce sea salt from open water and
blowing snow (Leaitch et al., 2018; Karl et al., 2019; Huang and Jaeglé et al., 2017; May et al., 2016) and significantly
decreased in summer (Willis et al., 2018; Leaitch et al., 2018). Likewise, the K-bearing particle fraction is higher in spring and
fall when long-range transports from biomass burning are enhanced (Yttri et al., 2014). Local dust particle emissions from the
bare soil surface and glacial outwash plains can be high in summer (Tobo et al., 2019), but only a few mineral dust particles
were observed (<1%) in our ambient aerosol samples in summer. This disparity is likely because mineral dust particles from
local sources are larger (1 µm) than the cutoff size of our samples.

### 3.3 Comparison of ambient aerosol and cloud residual samples

Cloud residual particles include CCN, INPs, and materials that dissolved or aggregated with the cloud droplets (Cziczo et al.,
2004). Investigations of cloud residuals provide information on what particles formed the cloud in the atmosphere. In particular,
INPs are a key to understanding the formations of ice crystals in Arctic mixed-phase clouds (Kanji et al., 2017; Murray et al.,
2021; Carlsen and David, 2022).

A comparison of the number fraction of each particle type between ambient aerosol and cloud residual samples showed that
mineral dust particles in cloud residual samples were almost 18 times higher than ambient aerosol samples in winter (Fig. 9).
Additionally, sea salt particles were approximately 2.3 times more abundant in cloud residual samples than in ambient aerosols
in all seasons. In contrast, sulfate particles were less abundant in cloud residual samples than in ambient aerosol samples (~0.7
times). In spring, the number fractions of K-bearing and carbonaceous particles in cloud residual samples were higher than in
ambient aerosol samples (1.8 and 1.5 times higher, respectively). In other seasons, the fractions showed opposite results.

To investigate the possible role of aerosol particles as CCN and INPs, cloud residual samples were classified according to
atmospheric temperature, assuming that those collected below 0℃ can contain residuals of ice crystals and supercooled cloud
droplets (Figs. 10 and 11). Comparing the number fraction of cloud residual samples collected below 0℃ with those above
0℃, mineral dust particles were higher in cloud residual samples below 0℃ (2%; Fig. S5) than those above 0℃ (0.1%). Cloud
residual samples collected below 0℃ had a higher sea salt particle fraction than those collected above 0℃ (41% and 16%,
respectively). Cloud residual samples collected below 0℃ had less sulfate particle fractions than those collected above 0℃



(42% and 72%, respectively; Fig. S5). There was no remarkable difference in K-bearing and carbonaceous particles between cloud residual samples collected below and above 0℃.

In addition to the average values through the entire campaign, ambient aerosol and cloud residual samples collected consecutively were analyzed (mostly within several hours) (Fig. 12). The comparison of these samples made it possible to

measure differences in the composition of samples collected from nearly identical air parcels and aerosol sources (Fig. 12). The results indicated that the mineral dust particle fraction in cloud residual samples collected at temperatures below 0℃ was 5.3 times higher than that in the ambient aerosol samples. Additionally, the mineral dust particle fraction in cloud residual samples increased with decreasing temperatures (Figs. 11 and 12). Sea salt particles were also abundant in the cloud residual samples, but there was no clear temperature dependence. The number fraction of K-bearing particles was higher in the ambient

aerosol samples than in the cloud residual samples and increased in both samples below 0℃. The fractions of sulfate particles in cloud residual samples were mostly lower than those in the ambient aerosol samples at all temperatures.

On the basis of the results of this study, we hypothesized possible pathways for each aerosol particle to become a cloud droplet. At temperatures above 0℃, the number fraction of ambient aerosol samples (Fig. S6) was similar to that of cloud residual samples (Figs. 10 and S5). This result suggests that most ambient aerosol particles, except those with low hygroscopicity,

acted as CCN and were activated into cloud droplets because of their large sizes. At temperatures below 0℃, cloud particles are a mixture of supercooled droplets and ice crystals that were formed through various freezing processes (e.g., immersion freezing, contact freezing, condensation freezing, and deposition ice nucleation (Kanji et al., 2017)). Meanwhile, supercooled water in droplets transfers to ice crystals through the Wegener–Bergeron–Findeisen process, i.e., the saturation vapor pressure of water is higher than that of ice, increasing the ice crystal fraction. The present study shows that the number fraction of

mineral dust particles in the residual samples increases at low temperatures (Figs. 12a and S5), suggesting that mineral dust particles act efficiently as INPs. The fraction of sea salt particles was higher in the cloud residual samples below and above 0℃ than in the ambient aerosol samples (Figs. 12b, S5, and S6). Sea salt will be discussed in more detail in the next section. It is unclear whether the observed cloud residuals consisting of sulfate, K-bearing, and carbonaceous particles were generated in cloud droplets or ice crystals below 0℃. However, given their hygroscopic nature, these particles were likely water droplets

in the temperature range of this study.

**3.4 Sea salt enrichment in cloud residual samples below 0℃**

Sea salt fractions in cloud residual samples collected below 0℃ were higher than those in ambient aerosol samples and those above 0℃ (Figs. 10 and S5). Additionally, the fraction of sea salt from cloud residual samples in summer, fall, and winter collected below 0℃ was the highest among all particle groups. Here, three possible processes will be discussed that may

contribute to the relatively high number of sea salt fractions in the cloud residual samples below 0℃. First, sea salt particles in the ice crystals contribute to the fractions in the cloud residual samples below 0℃. Second, similar to the high sea salt fractions in winter ambient aerosol samples, sea salt emissions increase with high wind speeds and low temperatures (Huang and Jaeglé, 2017) (Figs. 8 and S5). The ambient aerosol samples also show a higher sea salt fraction in samples collected below 0℃ (20 %) than in those above 0℃ (6%) (Fig. S6) because of an increase in sea salt aerosol emissions below 0℃ and a

decrease in other aerosol emissions (e.g., fewer sulfate emissions because of weak biological activity in the ocean at a low temperature). Finally, sea salt reacts with sulfate in cloud droplets, forming sodium sulfate that increases the sea salt fractions in cloud residual samples compared with those in the ambient aerosol samples. This possibility is suggested by a result that the sea salt fraction in the cloud residual samples collected above 0℃ was also higher than that in the ambient aerosol samples (16% and 6%, respectively; Figs. S5 and S6). These processes, more or less, could contribute to increased sea salt fractions in

cloud residual samples below 0℃.



### 3.5 Characterization of mineral dust particles for an implication of their INP activity

To further explore the role of mineral dust particles acting as potential INPs in cloud residual samples, we compared the composition of mineral dust particles from ambient aerosol samples with those from cloud residual samples collected below 0℃. The ternary ratios of Al, Si, and Fe are not significantly different between ambient aerosol and cloud residual samples (Fig. S7). Additionally, the average area equivalent diameters with standard deviations of mineral dust particles were $0.5 \pm 0.5$ µm for ambient aerosol samples and $0.4 \pm 0.2$ µm for cloud residual samples, showing no significant difference. These results indicate that the major composition and size of mineral dust particles in the ambient aerosol and cloud residual samples do not differ significantly.

The TEM measurements indicate that the substantial mineral dust particles are mixed with sea salt components (Fig. 2 and Table S1). Such mixing of mineral dust with sea salt has also been observed in samples collected from other Arctic campaigns (Adachi et al., 2021; Behrenfeldt et al., 2008; Geng et al., 2010). Therefore, using the characteristic elements of mineral dust and sea salt (Na, Mg, and Cl), we classified the mineral dust particles into two groups: with and without sea salt (Fig. 13). The ratios of Na, Mg, and Cl in these mineral dust particles are consistent with those of sea salt particles without mineral dust, confirming that these mineral dust particles are mixed with sea salt components. These plots also indicate that the mineral dust particles from cloud residual samples are mixed with sea salt more frequently than those from ambient aerosol samples, i.e., $43\pm7$ % of mineral dust particles in cloud residual samples contain sea salt, whereas only $18 \pm 5$% of the mineral dust particles in ambient aerosol samples contain sea salt.

When sea salt acts as a CCN, it becomes a droplet and may coagulate with mineral dust particles to form mixed particles. Mixed particles of mineral dust and sea salt have been detected in cloud residuals from various environments in previous measurements (e.g., Twohy and Poellot, 2005; Cziczo et al., 2004; Eriksen Hammer et al., 2018). Such mixed particles of mineral dust and sea salt could form a water layer on the surface of mineral dust particles and help develop ice crystals on these particles by immersion freezing (Kanji et al., 2017). Contrary, sea salt can lower the freezing point of the water layer and may adversely affect the ice formation of mineral dust particles. Further studies are required to confirm the hypothetical contribution of sea salt attached to mineral dust particles to INPs.

### 4. Conclusions

The composition and mixing state of individual ambient aerosol and cloud residual particles were analyzed in fine mode using a whole-air inlet, a PM$_{10}$ inlet, and a ground-based CVI inlet installed at the Zeppelin Observatory near Ny-Ålesund, an Arctic site, from March 2017 to February 2021. The number fractions among mineral dust, sea salt, K-bearing particles, sulfate, and carbonaceous particles showed seasonal variations, e.g., sulfate dominated in summer; sea salts increased in winter; and K-bearing particles increased in spring. The composition of cloud residual samples collected above 0℃ was similar to that of ambient aerosol particles, suggesting that fine-mode aerosol particles can commonly act as CCN in the Arctic low-level clouds. On the other hand, cloud residual samples collected below 0℃ have more mineral dust and sea salt particles and fewer sulfate than those collected above 0℃. Notably, the number fraction of mineral dust increased with decreasing temperature in cloud residual samples. These results suggest that the mineral dust particles act efficiently as INPs in the Arctic mixed-phase clouds. Sea salt particles may act as INPs or be present in supercooled water droplets. Sulfates may be less important for cloud seeds below 0℃ than cloud seeds above 0℃. Nearly half of the mineral dust particles collected below 0℃ were mixed with sea salt components. The influence of sea salt adhering to the surface of the mineral dust needs to be clarified. Further studies are recommended to (1) explore the contribution of coarse particles such as large mineral dust and biological particles, (2) quantitatively evaluate the increase in sea salt fractions in cloud residual samples below 0℃, (3) confirm the effect of sea salt attached to mineral dust, and (4) directly compare cloud residual particles with concurrent cloud phase measurements.



**Data availability**

TEM data for all individual particles and sample average are available at https://doi.org/10.5281/zenodo.7017936. Meteorological data are available at http://ebas-data.nilu.no.

**Author contributions**

KA conducted the TEM analysis and data processing. KA, YT, KM, GF, PZ, and RK executed the TEM sampling and field observations. MK and RK supervised the observations. KA prepared the manuscript with contributions from all coauthors.

**Competing interests**

Some authors are members of the editorial board of *Atmospheric Chemistry and Physics*. The peer-review process was guided 370 by an independent editor, and the authors have also no other competing interests to declare.

**Acknowledgments**

We are indebted to the staff of the Norwegian Polar Institute and Ove Hermansen from NILU for their cooperation and support for the observation at the Zeppelin Observatory. We appreciate the meteorological data provided by NILU. This work was supported by the Environment Research and Technology Development Fund (JPMEERF20205001, JPMEERF20202003, 375 JPMEERF20215003, and JPMEERF20172003) of the Environmental Restoration and Conservation Agency of Japan; the Global Environmental Research Coordination System of the Ministry of the Environment of Japan (MLIT1753); the Arctic Challenge for Sustainability (ArCS) project (JPMXD1300000000) and ArCS II project (JPMXD1420318865) of the Ministry of Education, Culture, Sports, Science and Technology (MEXT) of Japan; and the Japan Society for the Promotion of Science (JSPS) KAKENHI (grant numbers JP19H01972, JP19K21905, 19H04236, and JP19H04259). We would like to also thank for 380 financial support from Swedish Environmental Protection Agency (Naturvårdsverket), Knut and Alice Wallenberg Foundation (KWA) project "Arctic Climate Across Scales (ACAS)," and FORMAS funded project (# 2016-01427) "Interplay between water clouds and aerosols in the Arctic (IWCAA)."

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



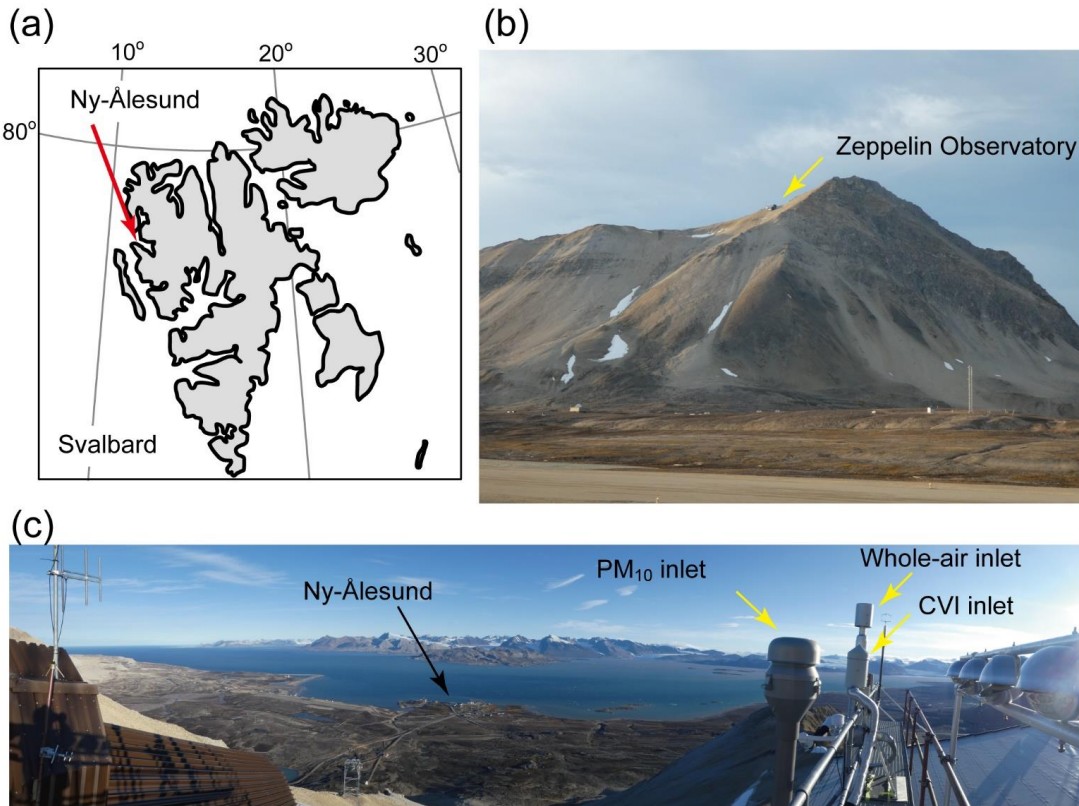


**Figure 1. Locations of Ny-Ålesund and the Zeppelin Observatory. (a) Map showing Ny-Ålesund, Svalbard, Norway. (b) Photo of Zeppelin Observatory taken in September 2017. (c) Panoramic view of Zeppelin Observatory showing three inlets used in this study and Ny-Ålesund. The PM$_{10}$ and whole-air inlets were used for ambient aerosol particle sampling. The CVI inlet was used for cloud residual particle sampling.**






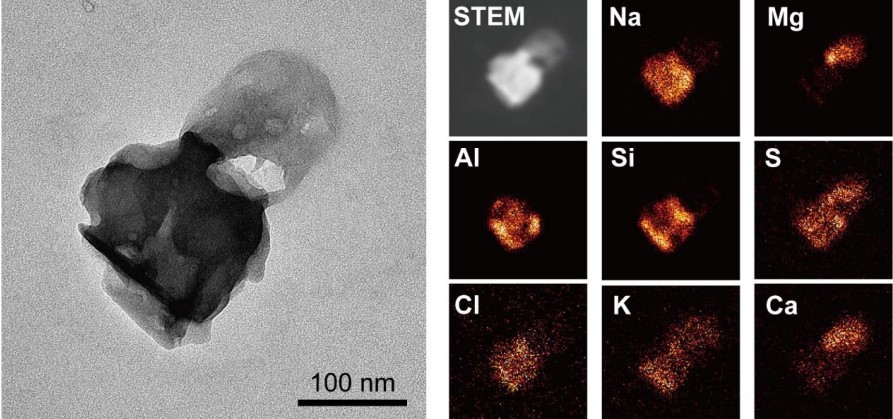

**Figure 2. TEM and elemental mapping images of a mineral dust particle mixed with sea salt. This particle was obtained from a cloud residual sample collected on November 12, 2018 (07:30).**






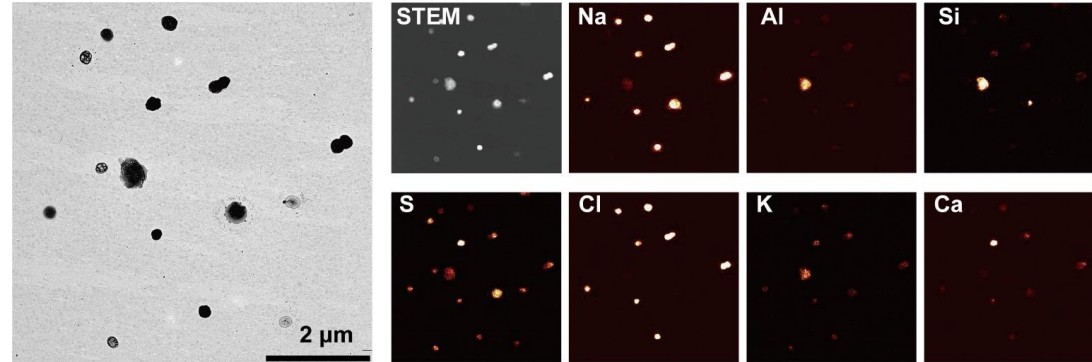

**Figure 3. TEM and elemental mapping images of a cloud residual sample collected on November 12, 2018 (07:30). The sampling temperature was −12℃.**

**(a)**

**(b)**

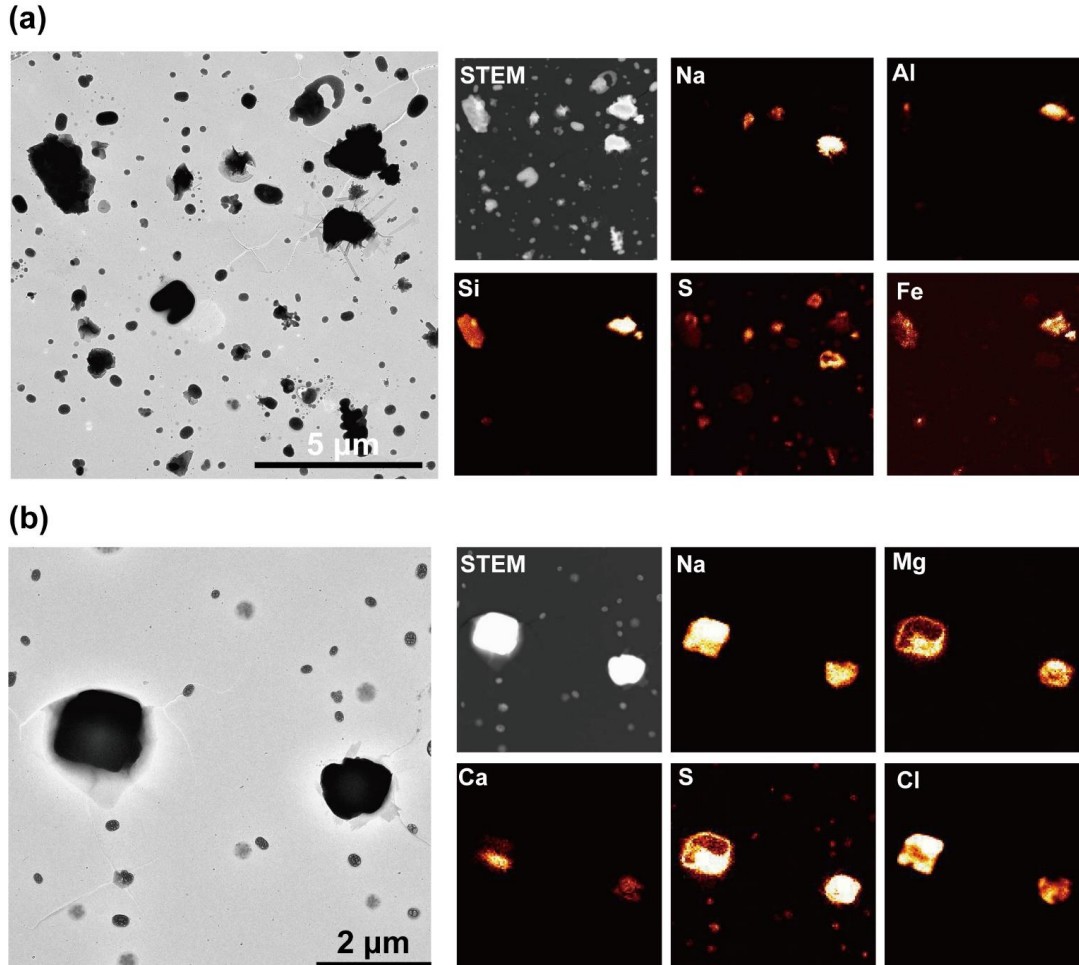

**Figure 4. TEM and elemental mapping images of an ambient aerosol sample with a relatively high (a) mineral dust particle fraction (10%) and (b) sea salt particle fraction (52%). Samples (a) and (b) were collected on July 13, 2018 (19:30) and August 30, 2018 (18:51), respectively. Mineral dust particles mainly consist of Al, Si, and Fe. Sea salt particles mainly consist of Na and Cl, with Mg and S on their surfaces.**



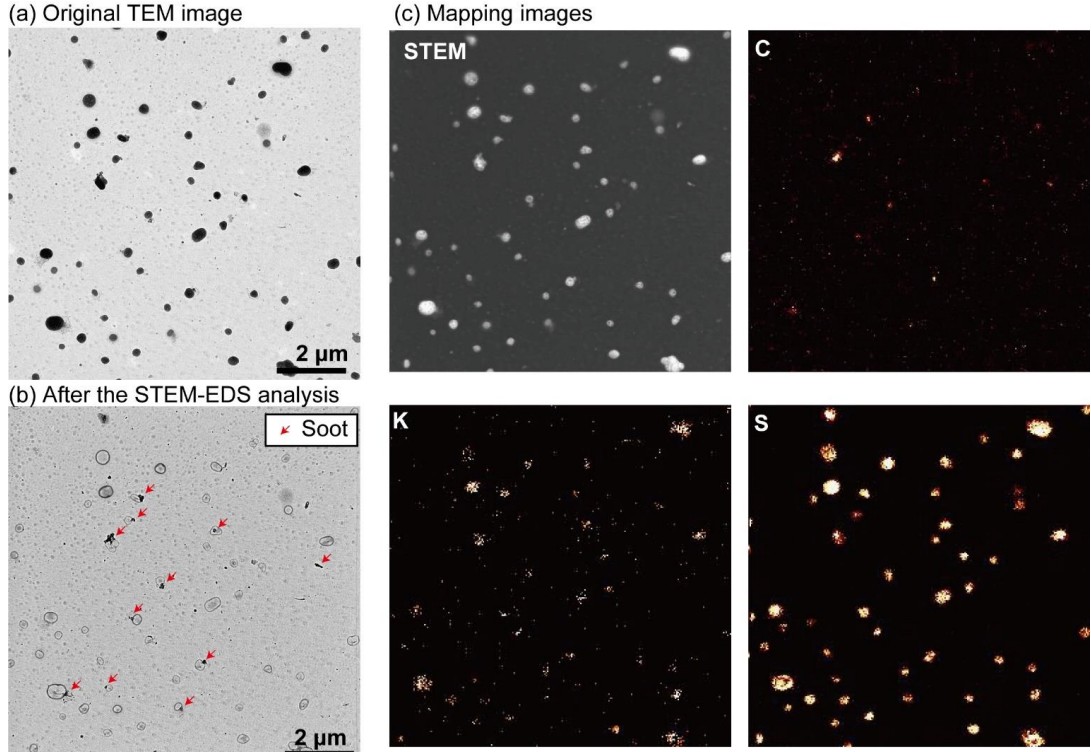

**Figure 5. TEM and elemental mapping images of an ambient aerosol sample with a relatively high K-bearing particle fraction (58%). (a) The TEM image shows the shapes of original particles before STEM-EDS mapping analysis. (b) The TEM image shows the same area after the STEM-EDS measurement. After removing potassium sulfates, internally mixed soot particles became apparent (red arrows). (c) STEM and elemental mapping images of C, K, and S. The distribution of C corresponds to soot particles. The distribution of S represents sulfates with and without K. The sample**
**was collected on April 29, 2019 (11:30).**



**(a) Sulfate dominant sample (Spring) (March 15 2018)**

**(b) Sulfate dominant sample (Summer) (August 10 2018)**

**(c) Organic dominant sample (September 8 2017)**

**Figure 6. TEM and elemental mapping images of ambient aerosol particles with relatively high sulfate particle fractions in (a) spring (76%), (b) summer (95%), and (c) particles with carbonaceous coatings. The samples were collected on (a) March 15, 2018 (12:00), (b) August 10, 2018 (12:00), and (c) September 8, 2017 (12:30). (a) and (b): Most particles are sulfate, with some containing Na and C (soot). (c): Organic matter has relatively low contrast (grey color) and coats sulfate particles (black). The distributions of C and S in the mapping images correspond to organic matter and sulfate particles, respectively.**






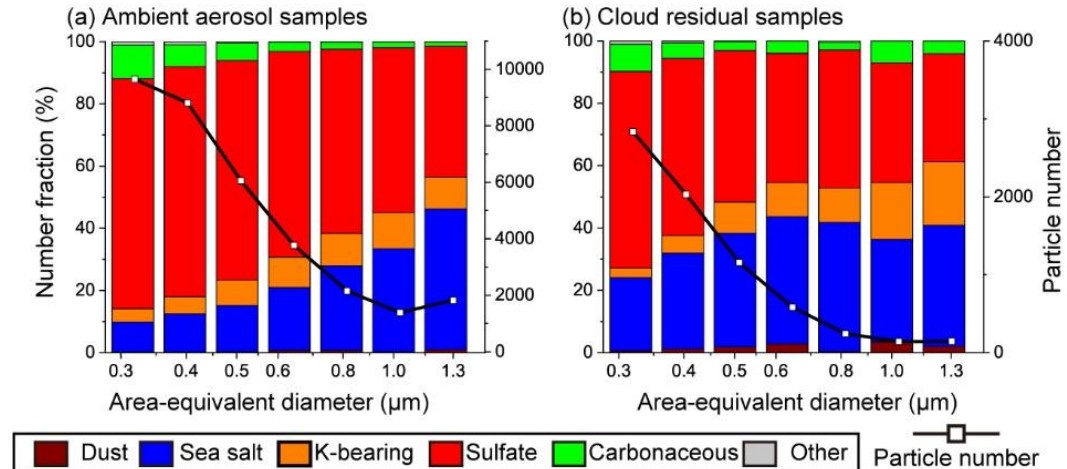

**Figure 7. Size-dependent number fractions of ambient aerosol and cloud residual particles and their number distributions. (a) Ambient aerosol particles (N = 33,652) and (b) cloud residual particles (N = 7138). The TEM samples used in (a) and (b) are 194 and 45, respectively.**

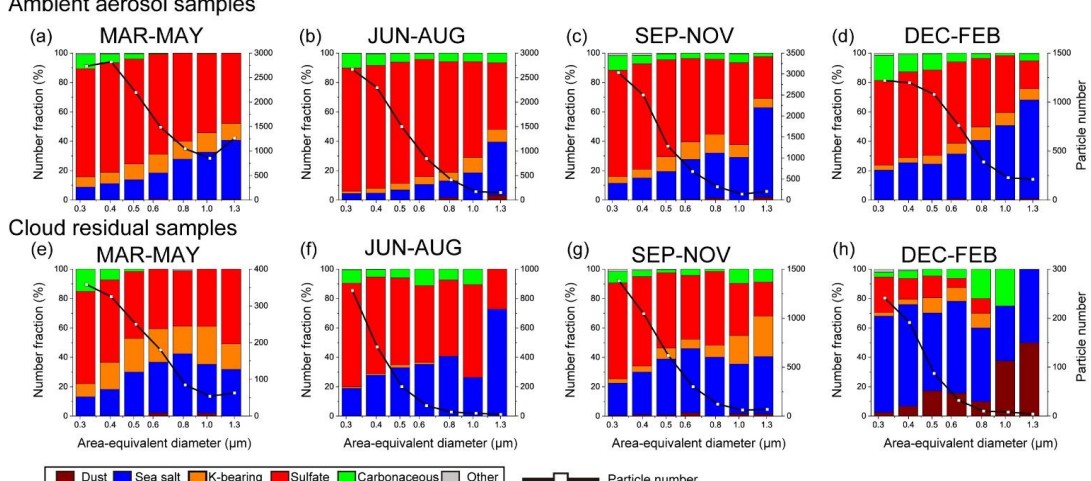

**Figure 8. Size-dependent number fractions and number distributions of ambient aerosol and cloud residual particles in each season. Ambient aerosol samples (a) from March to May, (b) from June to August, (c) from September to November, and (d) from December to February. Cloud residual samples (e) from March to May, (f) from June to August, (g) from September to November, and (h) from December to February. N = (a) 12,390, (b) 8037, (c) 8147, (d) 5078, (e) 1315, (f) 1652, (g) 3598, and (h) 573.**



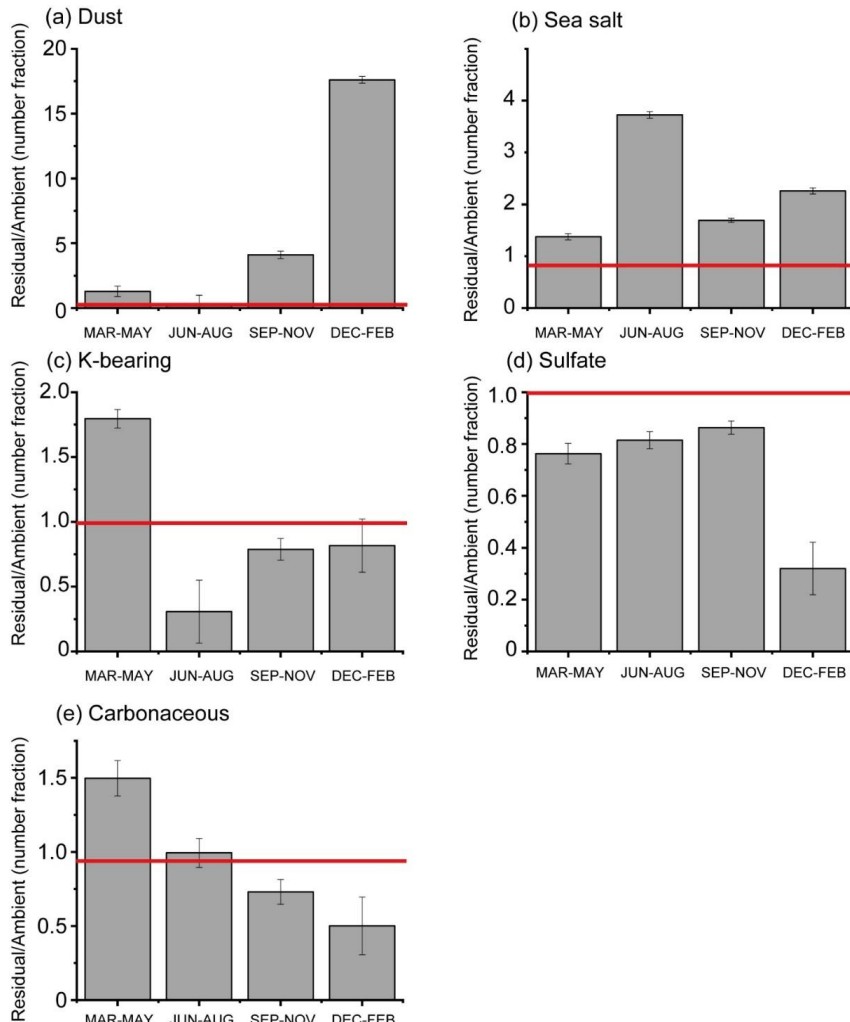

**Figure 9. Ratios of number fractions of each particle type between cloud residual and ambient aerosol samples for each season. (a) Mineral dust particles, (b) sea salt particles, (c) K-bearing particles, (d) sulfate particles, and (e) carbonaceous particles. Red lines indicate that the ratio is equal to 1. Errors are based on Poisson statistics.**





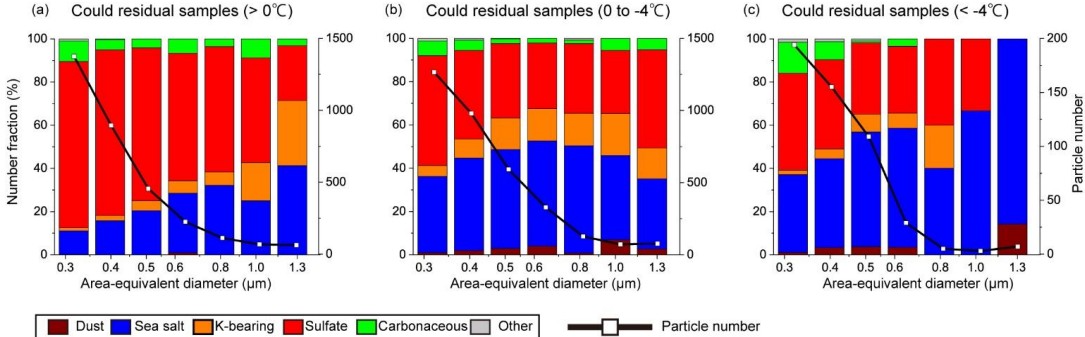

**Figure 10. Size-dependent number fractions and number distributions for cloud residual samples collected at (a) >0℃, (b) 0℃ to −4℃, and (c) <−4℃. N = (a) 3193, (b) 3443, and (c) 502.**



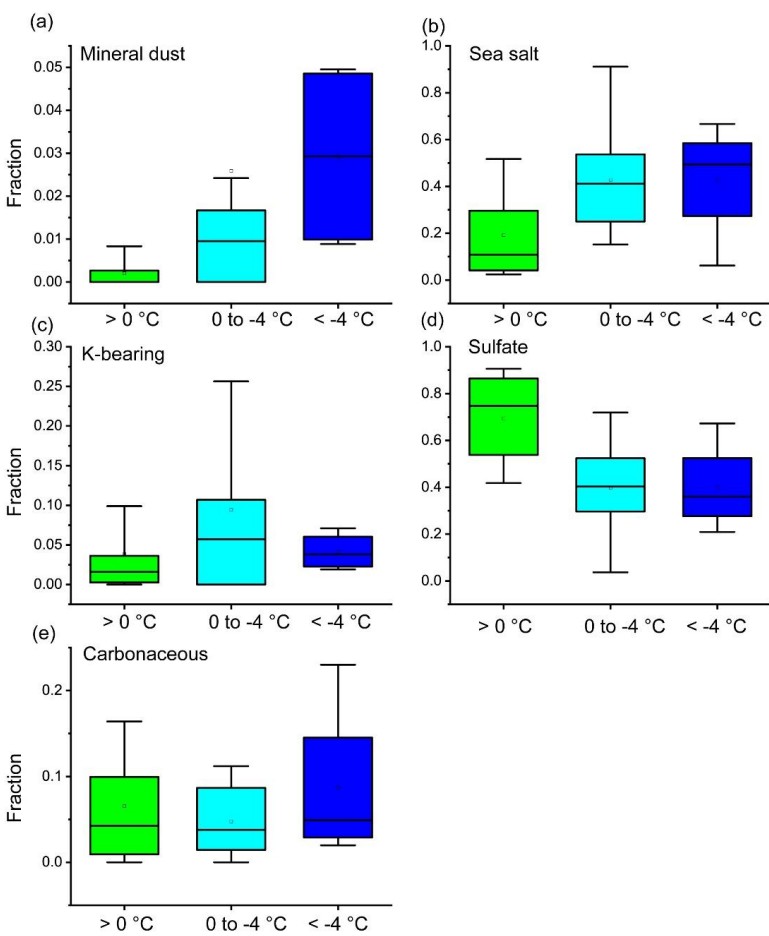

**Figure 11. Number fractions of each aerosol type in cloud residual particle samples collected at >0°C, 0°C to −4°C, and <−4°C. (a) Mineral dust particles, (b) sea salt particles, (c) K-bearing particles, (d) sulfate particles, and (e) carbonaceous particles. Sample numbers are 20, 25, and 4 for those collected at >0°C, 0°C to −4°C, and <−4°C, respectively. Wide boxes, whiskers, and squares indicate the 25th, 50th, and 75th percentile ranges; 10th–90th percentile ranges; and average values, respectively.**

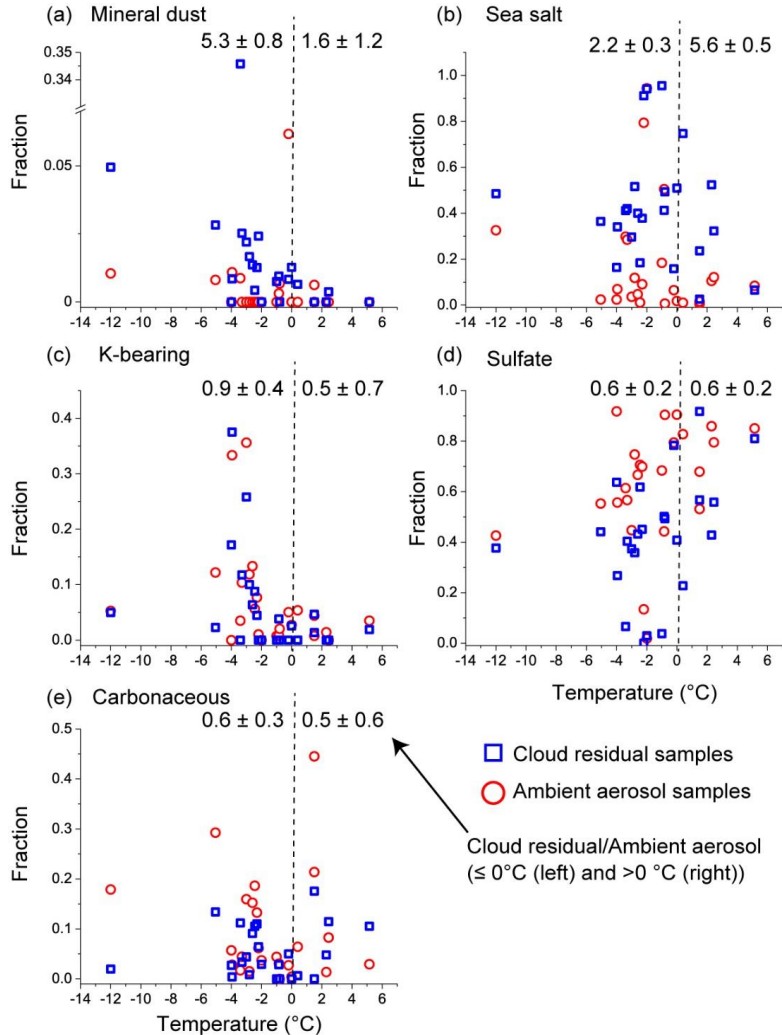

**Figure 12. Plots of number fractions of each aerosol particle type and the sampling temperature from 24 sets of ambient aerosol and cloud residual samples that were collected simultaneously. (a) Mineral dust particle fraction, (b) sea salt particle fractions, (c) K-bearing particle fraction, (d) sulfate particle fractions, and (e) carbonaceous particle fraction. Each plot indicates the average value of one to three samples collected within 24 h. Cloud residual samples and ambient aerosol samples are shown in blue squares and red circles, respectively. Values at the top of each panel indicate average fractions among cloud residual samples divided by those of ambient aerosol samples below 0°C (left) and above 0°C (right) with standard errors. Values greater than 1 indicate higher components in cloud residual samples than in ambient aerosol samples.**



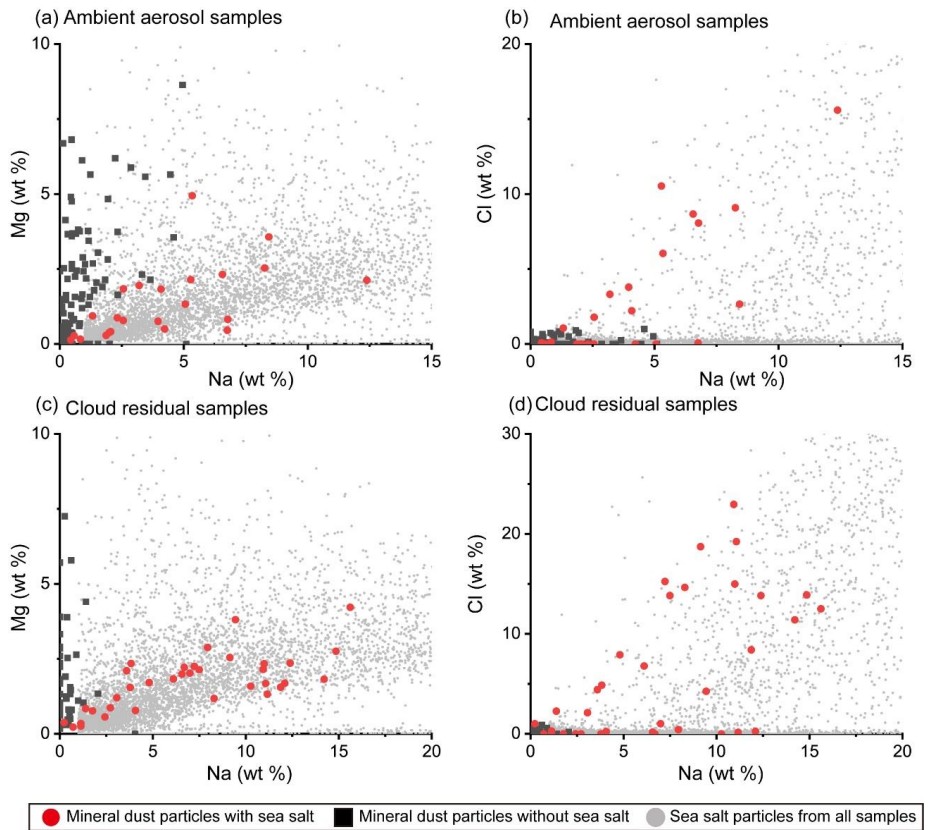

**Figure 13. Compositions of mineral dust particles with and without sea salt components in ambient aerosol and cloud residual samples. Plots of (a) Na and Mg and (b) Na and Cl (wt%) in mineral dust particles from ambient aerosol samples. Plots of (c) Na and Mg and (d) Na and Cl (wt%) in mineral dust particles from cloud residual samples. Red circles indicate mineral dust particles with sea salt components, defined as containing 0.5 > Mg/Na > 0.05, Cl > 1 wt%, or both. Black squares indicate mineral dust particles without sea salt components. Gray dots indicate all sea salt particles (Na > 1 wt%). N = 23 and 102 for mineral dust particles with and without sea salt in ambient aerosol samples, respectively. N = 35 and 47 for mineral dust particles with and without sea salt in cloud residual samples, respectively. N = 7697 for all sea salt particles.**





730 **Table 1. Detailed information of all samples used in this study**

| Sample classification | Sample subclassification | Sampling periods | No. of TEM samples | Analyzed particles | Temparature range (°C) |
|---|---|---|---|---|---|
| | | MM/DD YYYY–MM/DD YYYY | | | (Highest/Lowest) |
| Ambient aerosol samples | PM$_{10}$ inlet | 03/08 2017–03/28 2017 | 35 | 6231 | -6/-24 |
| | PM$_{10}$ inlet | 09/08 2017–09/12 2017 | 7 | 1039 | 6/1 |
| | PM$_{10}$ inlet | 03/12 2018–03/22 2018 | 20 | 4495 | -4/-16 |
| | PM$_{10}$ inlet | 08/02 2018–08/12 2018 | 16 | 2782 | 12/1 |
| | PM$_{10}$ inlet | 01/08 2019–01/13 2019 | 8 | 1559 | -9/-17 |
| | PM$_{10}$ inlet | 03/11 2019–03/13 2019 | 4 | 589 | -13/-16 |
| | PM$_{10}$ inlet | 07/29 2019–07/31 2019 | 4 | 887 | 8/3 |
| | PM$_{10}$ inlet | 11/10 2019–11/14 2019 | 6 | 655 | -4/-14 |
| | Whole-air inlet | 09/09 2017–02/08 2021 | 94 | 15415 | 9/-20 |
| Cloud residual samples | >0°C | 09/09 2017–09/11 2020 | 20 | 3193 | 6/0 |
| | 0 to -4°C | 09/19 2017–05/14 2019 | 21 | 3443 | 0/-4 |
| | <-4°C | 10/26 2017–11/12 2018 | 4 | 502 | -4/-12 |
| Total | | 03/08 2017–02/08 2021 | 239 | 40790 | 12/-24 |