# Peer review of "Composition and mixing state of Arctic aerosol and cloud residual particles from long-term single-particle observations at Zeppelin Observatory, Svalbard"

_Atmospheric Chemistry and Physics, 2022_

## Referee Comment (RC2)

**Comments to the ACPD**

Adachi et al. Composition and mixing state of Arctic aerosol and cloud residual particles from long-term single-particle observations at Zeppelin Observatory, Svalbard.

Reviewer's comments

The paper presents a study of ambient aerosol particles and residual cloud particles at the Zeppelin Observatory in Svalbard. The manuscript is interesting because the authors analyzed particle samples collected over four years in the Arctic, a region that can be greatly affected by global warming. The research focused solely on image analysis and a study of the elemental composition of atmospheric particles using TEM and STEM techniques coupled to an EDS detector. Although the manuscript is well structured and well written, some sections could be improved. For example, it could be very useful to present a general analysis of the meteorological conditions during each season, which could help to understand the seasonal differences in particle composition.

On the other hand, since similar electron microscopy studies of fine particles have already been reported (i.e., Weinbruch et al., 2012; Hara et al., 2003) the authors must place more emphasis on the large number of samples collected and analyzed over four years.

Introduction

- The authors should provide more information on studies on the elemental composition of fine particles. It would be important to provide a brief comparison with measurements of fine particle elemental composition.
- In línes 103-105: What does "TEM Samples Preset" mean? Does it refer to the nominal cutoff sizes of the sampler stages? It Would be important to indicate the 16 and 24 preset sizes in the samplers.

Methodology

- On line 106: it is important to indicate particle sizes obtained on TEM grids.
- The following sections (section 3.1.5) describe the elemental composition of the carbonaceous aerosol. Can the composition of the polymeric Formvar film affect the analysis of this type of particle (soot)?
- How is the effect of formvar composition on carbon quantification corrected with the EDS detector?
- In the article, a pixel/magnification ratio (100/6000x) is used to establish the particle size. In addition, particle size is measured using the equivalent diameter in area. Why didn't you use the physical diameter of the particles? Why didn't you use the microscope scale to establish particle size?
- Did you use the microscope scale to calibrate and establish particle size?

- Instead of establishing the diameter in terms of area, it might be better to make a relationship between the aerodynamic diameter (Da) and the physical diameter (Df). See the article by Wang, (1987).

Results

Is there any possible explanation for why temperature determines the composition of residual cloud samples? What role does relative humidity play on mineral dust and salt particles at temperatures below 0°C?

Consulted references

- Hara, K., Yamagata, S., Yamanouchi, T., Saton, K., Herber, A., Iwasaka, Y., Nagatani, M., and Nakata, H.: Mixing states of individual aerosol particles in spring Arctic troposphere during ASTAR 2000 campaign, J. Geophys. Res. Atmos., 108, https://doi.org/10.1029/2002jd002513, 2003.
- Wang, H. C.: Particle density correction for the aerodynamic particle sizer, Aerosol Sci. Technol., 6, https://doi.org/10.1080/02786828708959132, 1987.
- Weinbruch, S., Wiesemann, D., Ebert, M., Schütze, K., Kallenborn, R., and Ström, J.: Chemical composition and sources of aerosol particles at Zeppelin Mountain (Ny ålesund, Svalbard): An electron microscopy study, Atmos. Environ., 49, https://doi.org/10.1016/j.atmosenv.2011.12.008, 2012.

---

## Author Response (AR1)

Reviewer comments are in bold red.

Authors' comments are in black.

*Revised texts are in iatric.*

**Reviewer comment #1**

**Adachi et al. - Composition and mixing state of Arctic aerosol and cloud residual particles from long-term single-particle observations at Zeppelin Observatory, Svalbard.**

**As pointed out by the authors themselves right from the title and the introduction, this manuscript basically describes aerosol and cloud residual particle composition of background air over five years of sampling/measurements at the Zeppelin Observatory near Ny-Alesund. The great merit of the paper is, undoubtedly, that of providing, for the first time, long-term data and observations on aerosol and cloud residual particle properties and variability. This opens the horizons of understanding the processes underlying cloud chemistry and microphysics.**

**However, I believe that the quantitative observations reported and described in the results section deserves to be discussed more organically than in paragraph 3.3 (lines 292-305) and, in part, in paragraphs 3.2 and 3.4. The best would be to dedicate separate paragraphs to discussing the points that emerge in the previous ones - something like what you did in paragraph 3.5. This would provide a much clearer picture and a more direct link to the research perspectives reported in the final section.**

**Apart from this, it is good paper, well written and conceived in a clear and progressive way of explaining the topics little by little and in successive steps so that the questions that arise spontaneously in the reader can get their answer.**

Authors' comments to reviewer #1:

We appreciate reviewer #1's suggestion and supportive comments. The suggested sections and paragraphs have been reorganized as the following.

Section 3.2: We divided the section into two (sections 3.2 and 3.3) to focus on each issue (number fraction (3.2) and seasonality (3.3) of our samples).

Section 3.3 and 3.4 (now 3.4 and 3.5): We reorganized the section and made a clear distinction between the result and discussion.

Reviewer comment #2

Comments to the ACPD

Adachi et al. Composition and mixing state of Arctic aerosol and cloud residual particles from long-term single-particle observations at Zeppelin Observatory, Svalbard.

Reviewer's comments (RC)

RC1. The paper presents a study of ambient aerosol particles and residual cloud particles at the Zeppelin Observatory in Svalbard. The manuscript is interesting because the authors analyzed particle samples collected over four years in the Arctic, a region that can be greatly affected by global warming. The research focused solely on image analysis and a study of the elemental composition of atmospheric particles using TEM and STEM techniques coupled to an EDS detector. Although the manuscript is well structured and well written, some sections could be improved. For example, it could be very useful to present a general analysis of the meteorological conditions during each season, which could help to understand the seasonal differences in particle composition.

Authors' comments:

We appreciate the reviewer's supportive comments. We have added Table S2 that lists the average temperature, relative humidity, and wind direction for each season and cited it in Section 3.3.

Added: *The average temperatures with standard deviations were -10℃ ± 7, 2℃ ± 4, -4℃ ± 5, and -11℃ ± 6 for spring, summer, fall, and winter, respectively (Table S2), which are generally consistent with a long-term climatological data from 1993 to 2011 at Ny-Ålesund (Maturilli et at., 2013).*

RC2. On the other hand, since similar electron microscopy studies of fine particles have already been reported (i.e., Weinbruch et al., 2012; Hara et al., 2003) the authors must place more emphasis on the large number of samples collected and analyzed over four years.

Authors' comments:

We have revised the text to emphasize the large number of samples collected over a four-years observation period.

Original: *We measured the composition and mixing state of individual fine-mode particles using transmission electron microscopy.*

Revised: *We measured the composition and mixing state of individual fine-mode particles in*

*239 samples using transmission electron microscopy.*

Original: *Previous studies have measured individual particle compositions at Ny-Ålesund using samples collected during short-term campaigns. Instead, this study is the first attempt to provide a multiyear picture of aerosol and cloud residual particle properties and variability.*

Revised: *Previous studies have measured individual particle compositions at Ny-Ålesund using samples collected during short-term campaigns (e.g., Weinbruch et al., 2012). Instead, this study is the first attempt to provide a multiyear picture of aerosol and cloud residual particle properties and variability based on 239 TEM samples collected over a four-years observation period.*

**Introduction**

**RC3. The authors should provide more information on studies on the elemental composition of fine particles. It would be important to provide a brief comparison with measurements of fine particle elemental composition.**

Authors' comments:

We have mentioned additional information about fine particle studies at Ny-Ålesund in the Introduction section.

Added: *For example, Chi et al. (2015) found that sea salt particles with a range of coatings dominated fine particles in August, 2012 at Ny-Ålesund. Additionally, Gen et al. (2010) measured single-particle compositions of summertime aerosol particles in July, 2007 at Ny-Ålesund and found that reacted sea salt particles were abundant with particle sizes of 0.5–1.0 µm.*

**RC4. In lines 103-105: What does "TEM Samples Preset" mean? Does it refer to the nominal cutoff sizes of the sampler stages? It Would be important to indicate the 16 and 24 preset sizes in the samplers.**

Authors' comments:

The term "TEM Samples Preset" refers to a rotating disk that presets TEM grids. We have revised the sentences about the samplers. We have also revised the sentence about the cutoff sizes.

Original: *TEM samples were collected through three inlets (see Section 2.2 for inlet details) at the Zeppelin Observatory (Figs. 1c and S1) (Koike et al., 2019; Karlsson et al., 2021) using impactor samplers (AS-16W and AS-24W, Arios Inc., Tokyo, Japan).*

Revised: *TEM samples were collected through three inlets (see Section 2.2 for inlet details)*

*at the Zeppelin Observatory (Figs. 1c and S1) (Koike et al., 2019; Karlsson et al., 2021) using impactor samplers with rotation disks that load multiple TEM grids (AS-16W and AS-24W, Arios Inc., Tokyo, Japan).*

Original: *The AS-16W and AS-24W samplers have the same sampling conditions, except that the former has 16 TEM samples preset and the latter has 24 TEM samples preset.*
Revised: *The AS-16W and AS-24W samplers have the same sampling conditions and cutoff sizes, except that the former loads 16 TEM samples and the latter loads 24 TEM samples on the rotation disks.*

**Methodology**

**RC5. On line 106: it is important to indicate particle sizes obtained on TEM grids.**

Authors' comments:

We collected the fine-mode samples with aerodynamic diameters ranging from 0.1 to 0.7 µm (50% cutoff sizes).

Original: *All aerosol and cloud residual particles were collected on a 200-mesh Cu grid with a formvar carbon substrate (U1007, EM-Japan, Tokyo, Japan).*
Revised: *All aerosol and cloud residual particles were collected on a 200-mesh Cu grid with a formvar carbon substrate (U1007, EM-Japan, Tokyo, Japan) for both fine and coarse mode stages. This study focused on the fine-mode samples with aerodynamic diameters ranging from 0.1 to 0.7 µm (50% cutoff sizes).*

**RC6. The following sections (section 3.1.5) describe the elemental composition of the carbonaceous aerosol. Can the composition of the polymeric Formvar film affect the analysis of this type of particle (soot)?**

**How is the effect of formvar composition on carbon quantification corrected with the EDS detector?**

Authors' comments:

The substrate affects the measured weight % but not the particle classifications. The carbon signals from the substrate are included in the measured weight % values listed in Table S1. We mention the artifact in the footnote; "*Carbon can also be derived from the substrate.*" As TEM measures characteristic X-rays from samples and substrates excited by transmission electrons, the signals include both samples and substrates. The influence of the substrate depends on the particle density and thickness, which vary particle by particle, and it is difficult

to remove. However, they can be distinguished based on the EDS analysis because carbonaceous particles have a stronger carbon signal than the substrate (e.g., Fig. S4 in Adachi et al., 2016).

With the recognition of the artifact, we only used key elements to characterize particles (Fig. S2). Meanwhile, TEM images largely help in classifying particles accurately, and we used the images to confirm our classifications.

Soot particles were identified from both compositions (stronger carbon signal than the substrate) and their aggregated shapes (Fig. 5). Thus, the substrate did not affect the soot particle measurements shown in Fig. 5. We have added sentences to explain the above discussion.

Added: *The C and Cu signals in the spectra during EDS measurements can originate from the formvar carbon substrates and Cu-grids, respectively. Thus, we exclude Cu in our measurements, but we still show C weight % values that include C from both samples and substrate because they are important for identifying carbonaceous particles, which have stronger C signals than the substrate (Adachi et al. 2016).*

Added: *In addition to the compositions, TEM and STEM images of all particles were checked to confirm the particle classification.*

RC7. In the article, a pixel/magnification ratio (100/6000x) is used to establish the particle size. In addition, particle size is measured using the equivalent diameter in area. Why didn't you use the physical diameter of the particles? Why didn't you use the microscope scale to establish particle size? Did you use the microscope scale to calibrate and establish particle size?

Authors' comments:

We used an area-equivalent diameter, which is a type of physical (and geometric) diameter (DeCarlo et al., 2004). There are several geometric diameters with different definitions, such as Feret's and maximum diameters. We chose an area-equivalent diameter because the area-equivalent diameter shows a relationship with particle volumes (i.e., volume-equivalent diameter, Xu et al., 2020).

Please note that we did not use "a pixel/magnification ratio (100/6000x)" in the STEM measurements but a minimum pixel number of 100 at a magnitude of 6000 to identify a particle. The particle area and sizes are obtained using the STEM-EDS mode, in which the particle area is determined by the number of pixels that cover the particle.

The scales in STEM-EDS and TEM modes are regularly calibrated, and thus, both scales are identical.

We applied the STEM image to determine the same area for both composition and size measurements rather than using different particle definitions in TEM and STEM images. We have revised the sentence to clarify the meaning of magnification and pixel numbers.

Original: *For the STEM-EDS measurements, the minimum number of pixels for each particle at a magnification of 6000× in the STEM image was 100, resulting in the minimum particle size of 0.26 µm in area equivalent diameter.*

Revised: *For the STEM-EDS measurements, the magnification was fixed to 6000×, and the minimum number of pixels for each particle was 100, resulting in the minimum particle size of 0.26 µm in area equivalent diameter.*

**RC8. Instead of establishing the diameter in terms of area, it might be better to make a relationship between the aerodynamic diameter (Da) and the physical diameter (Df). See the article by Wang, (1987).**

Authors' comments:

We appreciate the suggestion and providing the interesting paper. However, to evaluate an aerodynamic diameter from a physical diameter, we also need to assume the particle density (Wang, 1987) and the particle volume for each particle. Therefore, we hesitate to use several assumptions to obtain the aerodynamic diameter and prefer to use an area-equivalent diameter in this manuscript. Please also note that the sampler collects particles with an aerodynamic diameter ranging from 0.1 to 0.7 µm (50% cutoff sizes), which will allow us to compare our results with those using aerodynamic diameter.

We explain a note to compare an area equivalent diameter with an aerodynamic diameter "*Note that for particles with a flat shape or low density, the area-equivalent diameter may be larger than the aerodynamic diameter because of differences in the definition of particle size.*"

**Results**

**RC9. Is there any possible explanation for why temperature determines the composition of residual cloud samples?**

Authors' comments:

At temperatures below 0℃, cloud particles are mixtures of supercooled droplets and ice crystals. At lower temperatures, the fraction of ice crystals can increase, and so do INPs in cloud residual samples. At temperatures above 0℃, the cloud residuals are liquid and can include more CCN than those below 0℃. Therefore, temperature could influence the number fractions of CCN and INPs, although the measurement of activated temperature in the cloud is required to

evaluate the fraction accurately. We explain the above discussion in the following sentences.

*All cloud residual particles above 0℃ are liquid droplets, whereas those below 0℃ are supercooled water droplets or ice crystals. Note that cloud droplets may have undergone colder temperatures than the sampling site, and the measured temperature may not be the same as the activated temperature for forming ice crystals (Carlsen and David, 2022).*
*To investigate the possible role of aerosol particles as CCN and INPs, cloud residual samples were classified according to atmospheric temperature, assuming that those collected below 0℃ can contain residuals of ice crystals and supercooled cloud droplets (Figs. 10 and 11).*

**RC 10. What role does relative humidity play on mineral dust and salt particles at temperatures below 0° C?**

Relative humidity (RH) data are added in the revised manuscript (Table S2). RH is important for sea salt particles because they deliquesce at ~75% RH, depending on temperature (Wise et al., 2012). Mineral dust particles do not deliquesce except for reacted Ca-bearing dust (Tobo et al., 2010), which were rarely observed in our samples. Mineral dust particles could be mixed with deliquesced sea salt particles. We have revised the following sentence to include the influence of RH.

*Original: When sea salt acts as a CCN, it becomes a droplet and may coagulate with mineral dust particles to form mixed particles.*
*Revised: When sea salt deliquesces at RH >~75%, depending on temperature (Wise et al., 2012), and acts as a CCN, it becomes a droplet and may coagulate with mineral dust particles to form mixed particles.*

References
Adachi, K., Moteki, N., Kondo, Y., and Igarashi, Y.: Mixing states of light-absorbing particles measured using a transmission electron microscope and a single-particle soot photometer in Tokyo, Japan, *J. Geophys. Res. Atmos.*, **121**, 10.1002/2016JD025153, 2016.
DeCarlo, P. F., Slowik, J. G., Worsnop, D. R., Davidovits, P., and Jimenez, J. L.: Particle morphology and density characterization by combined mobility and aerodynamic diameter measurements. Part 1: Theory, *Aerosol Sci. and Technol.*, **38**, 1185-1205, 10.1080/027868290903907, 2004.
Tobo, Y., Zhang, D., Matsuki, A., and Iwasaka, Y.: Asian dust particles converted into aqueous droplets under remote marine atmospheric conditions, *P. Natl. Acad. Sci. USA*, **107**,

17905–17910, 10.1073/pnas.1008235107, 2010.

Wise, M. E., Baustian, K. J., Koop, T., Freedman, M. A., Jensen, E. J., and Tolbert, M. A.: Depositional ice nucleation onto crystalline hydrated NaCl particles: a new mechanism for ice formation in the troposphere, *Atmos. Chem. Phys.*, **12**, 1121–1134, 10.5194/acp-12-1121-2012, 2012.

Xu, L., Fukushima, S., Sobanska, S., Murata, K., Naganuma, A., Liu, L., Wang, Y., Niu, H., Shi, Z., Kojima, T., Zhang, D., and Li, W.: Tracing the evolution of morphology and mixing state of soot particles along with the movement of an Asian dust storm, *Atmos. Chem. Phys.*, **20**, 14321–14332, 10.5194/acp-20-14321-2020, 2020.

---

## Author Response (AR2)

Editor comments are in red.
Authors' comments are in black.
*Revised texts are in Italic.*

Comments to the author:
I would like to thank the authors for incorporating the suggestions made by the reviewers. The revised version looks pretty good and it is almost ready for publication; however, I have the following additional and final comments before I can accept the manuscript.

Author Comments: We appreciate your handling our manuscript and providing suggestions to improve it. We have revised the manuscript, followed by the comments.

Minor/Technical Comments:
1.  Line 59: Add a reference after "spring"
We added references here.
*The Arctic atmosphere is heavily influenced by anthropogenic emissions from low- and mid-latitudes during winter and early spring **(Schmale et al., 2021; Willis et al., 2018)**.*

2. Line 60: Add a reference after "summer"
We added a reference here.
*In contrast, the influence of long-range transport is weakened in summer **(Willis et al., 2018)**.*

3. Line 135: Add the model and manufacturer of the used visibility sensor.
We added the information.
*The CVI was activated during cloud periods of < 1 km visibility as measured using a visibility sensor **(Belfort Instrument, USA, Model 6400)**.*

4. Line 258: I think "-10℃ ± 7, 2℃ ± 4, -4℃ ± 5, and -11℃ ± 6" could be changed to "-10 ± 7℃, ± 4℃, -4 ± 5℃, and -11 ± 6℃"
We revised the sentence as suggested.

***Original:*** *The average temperatures with standard deviations were **-10℃ ± 7, 2℃ ± 4, -4℃ ± 5, and -11℃ ± 6** for spring, summer, fall, and winter, respectively (Table S2),*

***Revised:*** *The average temperatures with standard deviations were **-10 ± 7℃, 2 ± 4℃, -4 ± 5℃, and -11 ± 6℃** for spring, summer, fall, and winter, respectively (Table S2),*

5. Line 343: It is unclear to me what the authors mean with "the substantial mineral dust particles"

We revised the sentence to indicate our meaning.

*Original: The TEM measurements indicate that the **substantial** mineral dust particles are mixed with sea salt components (Fig. 2 and Table S1).*

*Revised: The TEM measurements indicate that **many** mineral dust particles are mixed with sea salt components (Fig. 2 and Table S1).*

6. Figure 9. The red line is slightly misplaced in panels b and e

Thank you for pointing out the mistake. We have revised the figure.

[Figure]

7. Table 1. Given that two different sets of temperatures are shown in the Table, I suggest adding a few more details to make a clear distinction between the two sets of temperatures. We added footnotes to Table 1 and explained the temperatures.

**Table 1. Detailed information of all samples used in this study**

| Sample classification | Sample subclassification* | Sampling periods MM/DD YYYY–MM/DD YYYY | No. of TEM samples | Analyzed particles | Temparature range (°C)** (Highest/Lowest) |
|---|---|---|---|---|---|
| Ambient aerosol samples | $PM_{10}$ inlet | 03/08 2017–03/28 2017 | 35 | 6231 | -6/-24 |
|  | $PM_{10}$ inlet | 09/08 2017–09/12 2017 | 7 | 1039 | 6/1 |
|  | $PM_{10}$ inlet | 03/12 2018–03/22 2018 | 20 | 4495 | -4/-16 |
|  | $PM_{10}$ inlet | 08/02 2018–08/12 2018 | 16 | 2782 | 12/1 |
|  | $PM_{10}$ inlet | 01/08 2019–01/13 2019 | 8 | 1559 | -9/-17 |
|  | $PM_{10}$ inlet | 03/11 2019–03/13 2019 | 4 | 589 | -13/-16 |
|  | $PM_{10}$ inlet | 07/29 2019–07/31 2019 | 4 | 887 | 8/3 |
|  | $PM_{10}$ inlet | 11/10 2019–11/14 2019 | 6 | 655 | -4/-14 |
|  | Whole-air inlet | 09/09 2017–02/08 2021 | 94 | 15415 | 9/-20 |
| Cloud residual samples | >0°C | 09/09 2017–09/11 2020 | 20 | 3193 | 6/0 |
|  | 0 to -4°C | 09/19 2017–05/14 2019 | 21 | 3443 | 0/-4 |
|  | <-4°C | 10/26 2017–11/12 2018 | 4 | 502 | -4/-12 |
| Total |  | 03/08 2017–02/08 2021 | 239 | 40790 | 12/-24 |

* Samples were classified based on the inlets (ambient aerosol samples) and ambient air temperature when sampled (cloud residual samples).

** The highest and lowest temperatures during each sampling period.